# Achieving High Current Stability of Gated Carbon Nanotube Cold Cathode Electron Source Using IGBT Modulation for X-ray Source Application

**DOI:** 10.3390/nano12111882

**Published:** 2022-05-31

**Authors:** Yajie Guo, Junfan Wang, Baohong Li, Yu Zhang, Shaozhi Deng, Jun Chen

**Affiliations:** State Key Laboratory of Optoelectronic Materials and Technologies, Guangdong Province Key Laboratory of Display Material and Technology, School of Electronics and Information Technology, Sun Yat-sen University, Guangzhou 510275, China; guoyj37@mail2.sysu.edu.cn (Y.G.); wangjf65@mail2.sysu.edu.cn (J.W.); lbaoh@mail.sysu.edu.cn (B.L.); stszhyu@mail.sysu.edu.cn (Y.Z.); stsdsz@mail.sysu.edu.cn (S.D.)

**Keywords:** carbon nanotube, current stability, IGBT, X-ray source, current stability

## Abstract

The cold cathode X-ray source has potential application in the field of radiotherapy, which requires a stable dose. In this study, a gated carbon nanotube cold cathode electron gun with high current stability was developed by using Insulated Gate Bipolar Transistor (IGBT) modulation, and its application in X-ray source was explored. Carbon nanotube (CNTs) films were prepared directly on stainless steel substrate by chemical vapor deposition and assembled with control gate and focus electrodes to form an electron gun. A maximum cathode current of 200 μA and approximately 53% transmission rate was achieved. An IGBT was used to modulate and stabilize the cathode current. High stable cathode current with fluctuation less than 0.5% has been obtained for 50 min continuous operation. The electron gun was used in a transmission target X-ray source and a stable X-ray dose rate was obtained. Our study demonstrates the feasibility of achieving high current stability from a gated carbon nanotube cold cathode electron source using IGBT modulation for X-ray source application.

## 1. Introduction

X-ray source has important applications in the fields of industrial inspection, medical diagnosis and cancer radiotherapy, etc. [1,2,3]. The traditional thermionic cathode X-ray sources have the disadvantages of slow response, bulky volume, high-power consumption and short lifetime even though they are widely used nowadays [4]. The cold cathode X-ray sources have been developed to solve those problems. By using cold cathode, the X-ray source can achieve instant turn-on and off, small size, low power consumption and long lifetime. In addition, it is easy to realize miniaturized or micro-focus X-ray sources. Cold cathode X-ray sources have thus become a research hotspot in the field [5,6,7,8,9].

Since 2001, carbon nanotubes (CNTs) cold cathode has been extensively studied for X-ray sources application due to its ultra-high aspect ratio and excellent electrical characteristics [10]. H. Sugie et al. first fabricated a prototype X-ray tube with carbon nanotube cold cathode prepared on cobalt coated tungsten wire. X- ray imaging was achieved under 1.5 μA of cathode current and anode voltage of 60 kV and the lifetime of more than 60 min was obtained [11]. After that, extensive studies have been carried out for miniature, pulsed or micro-focus CNT cold cathode X-ray source [12,13,14,15,16,17,18,19,20]. For example, Park et al. has developed a fully vacuum-sealed miniature X-ray tube with CNT emitters for portable dental X-ray imaging systems [18]. The operational voltage of 65 kV and a current of 3 mA has been achieved. Yue et al. fabricated a pulsed CNT cold cathode X-ray source, and a total emission current of 28 mA was obtained from a 0.2 cm^2^ area CNT cathode [19]. The CNT cold cathode X-ray source had imaged a human extremity with pulsed anode voltage of 14 kV and can potentially be applied for industrial and medical applications. Liu et al. reported a CNT cold cathode microfocus field emission X-ray tube [20]. X-ray imaging was achieved at anode current of 0.1 mA and at the peak anode voltage of 40 kV.

Some applications of CNT cold cathode X-ray source in medical X-ray system have been demonstrated. Cao et al. reported a Computed Tomography (CT) scanner using a compact CNT X-ray tube [21]. The system could achieve 100 μm spatial resolution and 20 msec temporal resolution, which has been adopted in cardiac imaging of free-breathing mice. Hadsell et al. proposed a compact X-ray-based Medical Radiologic Technology (MRT) system that employed CNT cold cathode [22]. A dose rate of 1.54 mGy/s has been obtained under 50 kV anode voltage and 0.5 mA anode current, which demonstrated the feasibility of compact MRT system with a high dose rate. Zhang et al. used CNT cold cathode X-ray source to develop a multi-beam X-ray imaging system [23]. The device can provide a tube current of 0.1–1 mA at 40 kV and realize fast data acquisition for tomographic imaging. Cheng et al. has reported a dynamic radiography system with a CNT cold cathode X-ray source [24]. The system achieved ultra-fast temporal resolution and has potential applications for dynamic X-ray imaging, which is promising for biomedical research.

Application imposes stability requirement on CNT cold cathode X-ray source [25]. For example, in radiotherapy application, high dose stability over a long period of (up to half an hour) continuous operation is essential, and thus high stable operation of cold cathode is a must [26,27,28]. Several approaches have been studied for achieving high stable emission from gated CNT cold cathode electron source for X-ray source application. Some researchers optimized the preparation method or adopted a post-treatment process [29,30]. Xiomara et al. put forward an electrophoretic process to fabricate composite CNT films with controlled nanotube orientation and surface density, and enhanced adhesion [29]. The cathodes have significantly enhanced macroscopic field emission current density and long-term stability under high operating voltages. The peak emission current in the pulsed mode was fixed at 3 mA and an average peak current of 3.0 ± 0.1 mA was obtained under the modulation of gate voltage. The application of this CNT electron source for X-ray imaging is also demonstrated. Ji et al. has proposed a post-processing technology of electrical aging for CNT cathode prepared using resist-assisted patterning (RAP) and direct current (DC) plasma-enhanced CVD [30]. An anode current of 5.27 mA and a current stability of 2.4% were obtained at 1400 V gate voltage. X-ray images were obtained under 65 kV anode voltage. Although others use electronic circuit (active or passive) to stabilize the emission current [31,32], Sun et al. have adopted a passive ballast resistor under CNT to improve reliability and current stability [31]. Moreover, Kang et al. has reported using an advanced active-current control (ACC) circuit to increase the stability of emission current in a CNT cold cathode X-ray source [32]. Despite the above-mentioned progresses, further study is needed for achieving high stable current and high dose stability of X-ray source using active circuit.

In this work, a gated CNT cold cathode electron gun has been fabricated for high current stability X-ray source application. CNTs have been prepared by CVD. The I-V characteristic curve and electron transmission have been tested under a high-vacuum system. Current stability has been studied and a high voltage Insulated Gate Bipolar Transistor (IGBT) has been used to suppress the current instability. Finally, the fabricated gated CNT cold cathode electron gun was used in a transmission target X-ray source and the dose rate stability was measured.

## 2. Device Fabrication and Cathode Preparation

Figure 1 shows the structure and the measurement circuit of a gated CNTs cold cathode electron gun. The electron gun is composed of CNTs cold cathode, control gate and focus electrodes. The cathode comprises CNTs thin film prepared on round stainless steel (SS 304) substrate of 1.6 mm in diameter and 0.2 mm in thickness. The gate is a molybdenum mesh fixed on a stainless steel (SS 304) support. The 2 mm diameter mesh is composed of round holes with a diameter of 90 µm and spacing of 30 µm. The focus electrode is a cylinder with a height (d) of 6 mm and inner diameter of 11 mm. The cathode, gate support and focus electrode were installed on a circular ceramic base. The gap between the cathode and the mesh gate is approximately 100 μm, which is defined by a thin metal ring.

For the measurement of the current-gate voltage (I-V) curve and current stability of gated CNTs cold cathode electron gun, a phosphor screen was used as the anode. The distance (D) between the ceramic base and phosphor screen is 11 mm. The current-voltage characteristics of the electron gun are measured in a high vacuum chamber with a base pressure of 2.0 × 10^−5^ Pa. The anode is connected to a high voltage power supply (TD2200, Teslaman. Tech Co., Ltd., Dalian, China), capable of biasing up to a maximum voltage of 50 kV. The gate and focus voltage were supplied by individual digital manual-mode voltage sources. Anode current (I_a_), gate current (I_g_), focus electrode current (I_f_) and cathode current (I_c_) versus gate voltage (V_g_) were recorded by amperemeters.

IGBT is a voltage-driven power semiconductor device combing BJT (Bipolar Junction Transistor) and MOSFET (Metal Oxide Semiconductor Field Effect Transistor) [33]. It can form a channel to drift charge carriers which can achieve a stable current by adding a fixed positive gate voltage within a certain range [33,34]. In this study, an IGBT (SP25N135T, Xiner, Shenzhen, China) was incorporated into the cathode circuit as shown in Figure 1. The collector of IGBT was connected in series with the cathode. The source of IGBT was grounded through a digital amperemeter. The gate of IGBT was linked to a source metre (Keithley 2450, Tektronix, Beaverton, OR, USA), which can provide a highly stable voltage output.

The CNTs were prepared by thermal chemical vapor deposition (TCVD) [35]. First, the 0.5 nm thickness Fe thin film was pre-deposited by sputtering on the SS 304 stainless steel substrate acting as the catalyst. Then, the substrate was placed in the centre of a quartz tube furnace. Secondly, the tube furnace was pumped to 1.5 × 10^−3^ Pa by the mechanical pump and turbo pump. Then mixture gases of hydrogen and argon were continuously introduced while the tube was heated up to 650 °C in 60 min. Subsequently, the reaction gases of acetylene and hydrogen were let in and the CNTs were grown on the substrate. After 20 min, the reaction gas was stopped, and the mixture gas of hydrogen and argon was resumed until the temperature lowered to room temperature.

The morphology the prepared CNTs were characterized by SEM (SUPRA™60, Zeiss, Oberkochen, Germany). The crystalline structure was studied by HRTEM (Titan G2 300 KV, FEI, Columbus, NJ, USA). In addition, Raman spectra were obtained by Raman spectroscopy (FLSP920, Edinburgh Instruments, Edinburgh, UK) with a 532 nm Ar ion laser source.

For X-ray source device, a transmission target made of 1070 nm molybdenum thin film on the quartz substrate was used as the anode [36]. The distance between the ceramic base and molybdenum anode is about 50 mm. The X-ray source was measured in the vacuum chamber with a beryllium window. The X-ray dose rate was measured by a dose detector (Magic Max, IBA, Göttingen, Germany) which is installed before the beryllium window. A cadmium telluride (Cd-Te) X-ray energy spectrum detector (X-123SDD, AMPTEK, Inc, Bedford, MA, USA) was used to measure the X-ray energy spectrum. Finally, the X-ray imaging was carried out by using a flat panel X-ray detector (Xineos-1515, Thousand Oaks, CA, USA).

## 3. Results and Discussion

### 3.1. Materials Characterization

Figure 2a,b shows the overview morphology of the prepared CNTs under different magnifications. The prepared CNTs are dense and have randomly aligned spaghetti-like morphology. The diameter of CNTs is about 20–30 nm and the length of CNTs reaches several tens of microns. Figure 2c shows the cross-sectional morphology of prepared CNTs. The CNTs twisted and tangled randomly and some CNTs protrude 10–20 µm above other CNTs. Figure 2d shows magnified SEM of one protruding CNTs.

The HRTEM of the prepared CNTs was shown in Figure 3a, which shows a multi-wall structure. Figure 3b shows the Raman spectrum. The Raman shift at 1349 cm^−1^, 1587 cm^−1^ corresponding to the D peak and G peak of CNTs. The intensity ratio of I_G_/I_D_ is about 0.57, indicating a relative-high defect level in the graphene structure of prepared CNTs. Weak sideband can also be found on the right side of G peak, which represents lattice defect of prepared CNTs [37,38].

### 3.2. Field Emisison Characteristics and Stability of Gated CNT Electron Gun

The field emission current-voltage (I−V) characteristics of the CNT cathode electron source are measured in a vacuum chamber. The anode current (I_a_), gate current (I_g_), focus electrode current (I_f_) and cathode current (I_c_) versus gate voltage (V_g_) characteristics of the electron gun were measured under 6 kV anode voltage and 0 V focus voltage, as shown in Figure 4a. Maximum current of 200 μA has been achieved at gate voltage of 1450 V, which corresponding to a current density of 9.55 mA/cm^2^ at the cathode. The field emission current was usually described by the Fowler–Nordheim (F–N) equation [39]:(1)$$I=\mathrm{AS}\left(\frac{{\beta V}^{2}}{\Phi h^{2}}\right)\exp\left(-{B\Phi }^{\frac{3}{2}}h/\beta V\right)$$
where A and B are two constants, Φ is the work function of emitter, β is field enhancement factor, S is the emission area, h is the distance between cathode and gate and V is applied voltage. The inset of Figure 4a shows the corresponding F–N plot and the field emission image. A deviation from linear relation was observed, which can be attributed to space charge effect according to early studies [40,41].

The electron transmission rate was calculated using I_anode_/I_cathode_. The transmission rate results under different gate voltage are shown in the Figure 4b and the assembled electron gun device is also shown in the inset of Figure 4b. The electron transmission rates are around 53% under different gate voltages.

We further studied the current stability of the CNT cold cathode electron source with and without IGBT regulation. In the measurement, the anode voltage was fixed as 4 kV. Figure 5a shows the cathode current stability of CNT cold cathode electron source under different current levels without IGBT modulation.

The current fluctuation φ was calculated by [42]:(2)$$\varphi =\sum_{i=1}^{n}\frac{\left|I_{i}-I_{\mathrm{average}}\right|}{n.I_{\mathrm{average}}}$$
where I_i_ is the cathode current at each moment and I_average_ is the average current. The calculated current fluctuations are 3.97%, 3.91% and 7.34% for gate voltage of 1.0 kV, 1.1 kV and 1.2 kV, respectively. The current instability of CNT electron source mainly resulted from surface adsorbates of CNT cold cathode [43].

Furthermore, the cathode current stability was measured when the IGBT was connected to the cathode to modulate the emission current. The gate voltage of the CNT electron source was set to 1.2 kV and the gate voltage of IGBT was tuned. Figure 5b shows the results when the IGBT gate voltage is fixed at 5.3 V, 5.4 V and 5.5 V. We found that the emission current can be effectively modulated by the IGBT gate voltage and current of 15.3 μA, 28.5 μA and 49.3 μA were obtained under the different IGBT gate voltages. The current fluctuations were calculated using equation 2 and the obtained values are 0.22%, 0.45% and 0.24% when the IGBT gate voltage is 5.3 V, 5.4 V and 5.5 V, respectively.

The electrical characteristics of the IGBT was measured, as shown in Figure 6. The collector currents were ~2.21 μA, ~4.14 μA, ~7.72 μA, ~14.5 μA, ~27.6 μA, ~53.2 μA and ~96.8 μA when the IGBT gate voltage was set as 5.0 V, 5.1 V, 5.2 V, 5.3 V, 5.4 V, 5.5 V and 5.6 V, respectively. Clearly, the IGBT works in the saturation region which has high stability current. When the emission current from CNT cathode fluctuates, the IGBT will limited the current in the circuit, lowering the fluctuation in the emission current.

Table 1 compares the current stability of the CNT electron source reported in references. The results indicate the current stability achieved in our work are better compared with the referred data. Using IGBT modulation is an effective approach to overcome the instability in CNT cold cathode and realize high current stability.

### 3.3. Application in X-ray Source

The fabricated CNT cold cathode electron gun was assembled with a transmission anode to form an X-ray source. The X-ray emission of the device was measured in a high vacuum chamber with a beryllium window. The gate voltage of IGBT was set to 5.5 V and the anode voltage is set to 45 kV. The dose rate data were collected by a dose detector within 100 s;the result is shown in Figure 7. A stable dose rate of about 15.5 µGy/s was achieved, and the high stability proved the effect of IGBT modulation.

The energy spectrum was recorded under 50 kV anode voltage by the Cd-Te X-ray energy spectrum detector, as shown in Figure 8. The two peaks at 17.48 and 19.58 keV in the X-ray spectrum correspond to characteristic emissions of the K-shell of the Mo target [48].

The X-ray imaging properties of the X-ray source were also studied. The X-ray focal spot was measured following the European standard (EN 12543-5), where the resolution is obtained based on the line profiles of the transmitted X-ray intensity of a 1 mm diameter tungsten wire phantom in two orthogonal directions [20]. To satisfy the conditions set by this standard, the wire and flat panel X-ray detector were placed, respectively, 15 and 20 cm apart from the X-ray source, horizontally. The X-ray projection image of the tungsten wire phantom is shown in Figure 9a. The horizontal and vertical size of the X-ray focal spot was calculated to be about 473 µm and 521 µm from the line intensity profiles of the cross wires shown in Figure 9b. The focal spot demagnification factor is approximately 3.2, which is calculated from the ratio of the cathode size (160 µm) and the true focal spot diameter. X-ray imaging results of a line pair card was presented in Figure 9d and up to 3.55 line pairs/mm was clearly resolved, indicating that the imaging resolution of CNT cold cathode X-ray source can reach to approximately 140 µm. The integrated circuit chips were also clearly imaged as shown in Figure 9d.

## 4. Conclusions

A gated carbon nanotube cold cathode electron gun with high current stability was achieved using IGBT modulation. Maximum cathode current of 200 μA and approximately 53% transmission has been achieved. High stable cathode current with less than 0.5% fluctuation has been obtained for 50 min continuous operation by using IGBT modulation. A transmission target X-ray source was fabricated with the electron gun. Stable X-ray dose rate and clear X-ray imaging were obtained. The results demonstrate that IGBT modulation is an effective way to achieve high current stability of gated carbon nanotube cold cathode electron source for X-ray source application.

## Figures and Tables

**Figure 1 nanomaterials-12-01882-f001:**
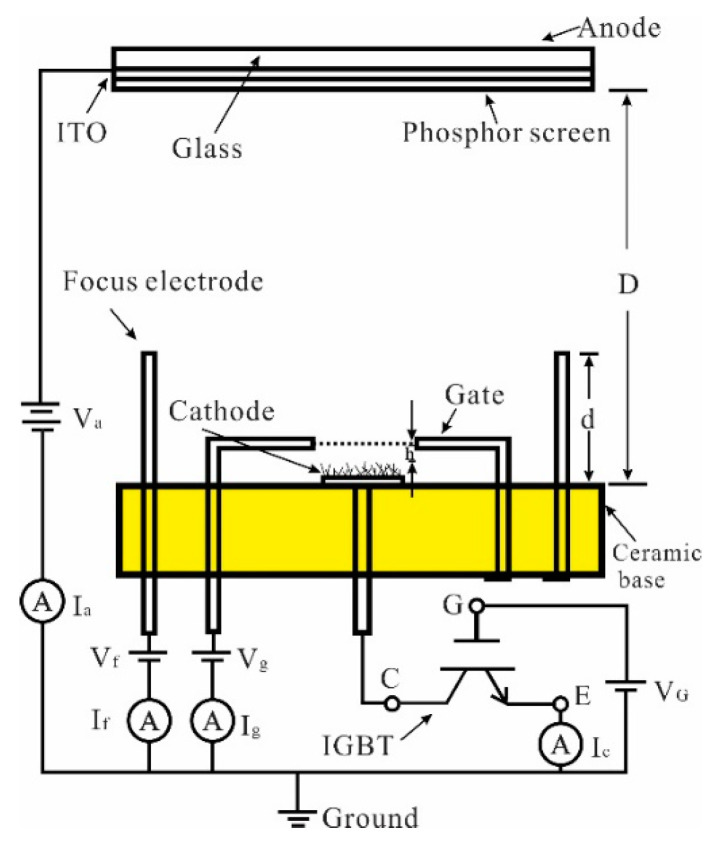
The structure and measurement circuit of CNTs cold cathode electron gun.

**Figure 2 nanomaterials-12-01882-f002:**
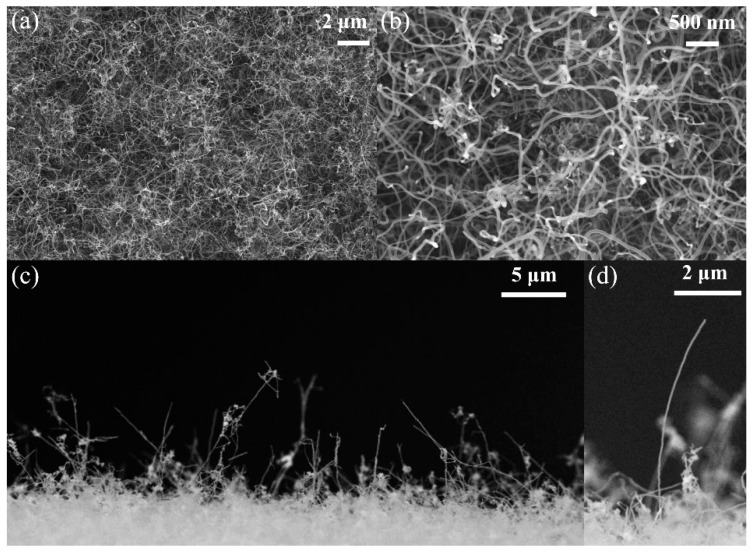
SEM images of the prepared CNTs. (**a**,**b**) The overview morphology under different magnifications. (**c**) The cross-sectional view of grown CNTs. (**d**) SEM image of one protruding CNT.

**Figure 3 nanomaterials-12-01882-f003:**
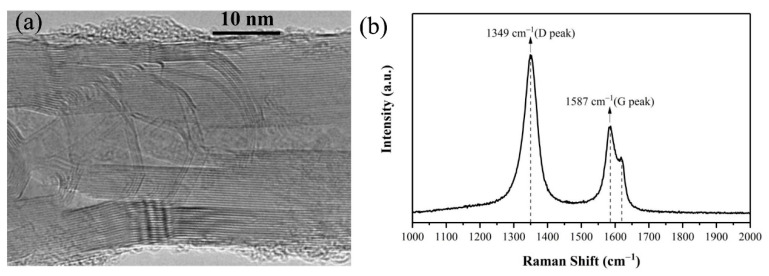
(**a**) HRTEM image and (**b**) Raman spectrum of the prepared CNTs.

**Figure 4 nanomaterials-12-01882-f004:**
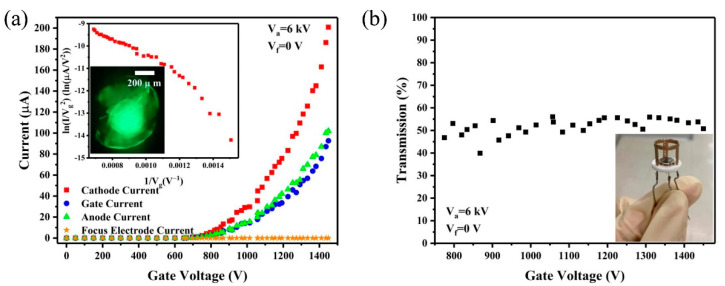
(**a**) I−V curves of the CNT cold cathode electron gun (Inset: corresponding F–N plot and emission image obtained from phosphor screen) (**b**) The gate electron transmission under different gate voltage. (V_a_ = 6 kV, V_f_ = 0 V). Inset shows the picture of the gated CNT cold cathode electron gun.

**Figure 5 nanomaterials-12-01882-f005:**
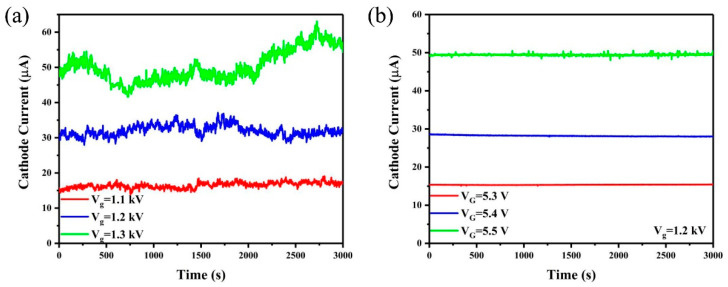
The cathode current stability of CNT cold cathode electron source (**a**) without and (**b**) with IGBT modulation.

**Figure 6 nanomaterials-12-01882-f006:**
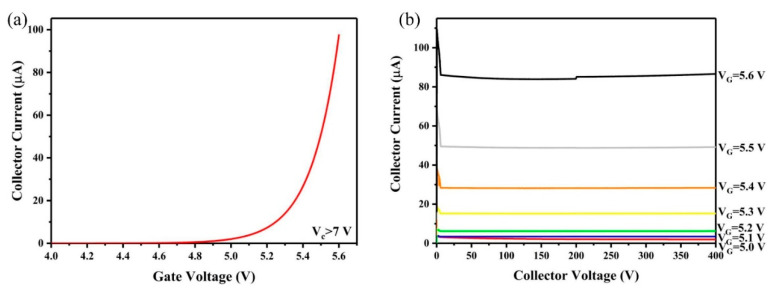
(**a**) Transfer and (**b**) output characteristics of the IGBT used in our study.

**Figure 7 nanomaterials-12-01882-f007:**
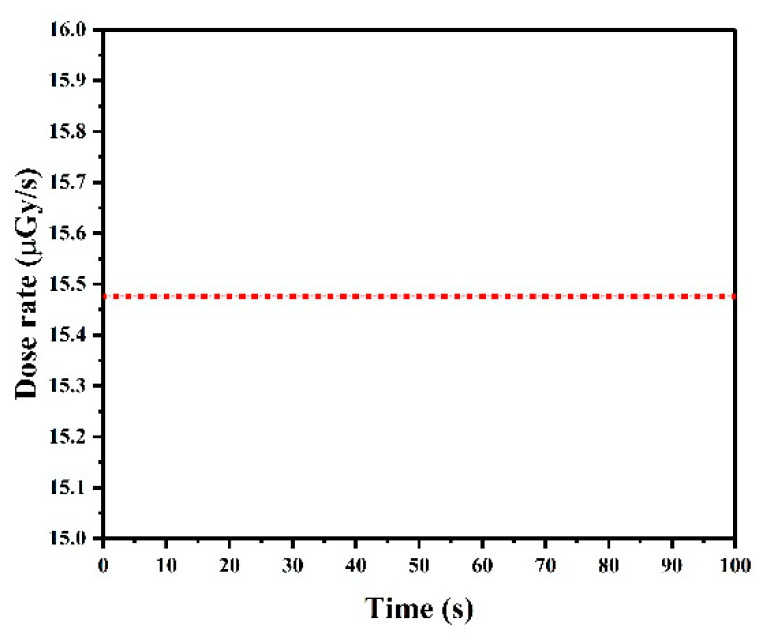
The X-ray dose rate of gated CNTs X-ray source at V_a_ = 45 kV.

**Figure 8 nanomaterials-12-01882-f008:**
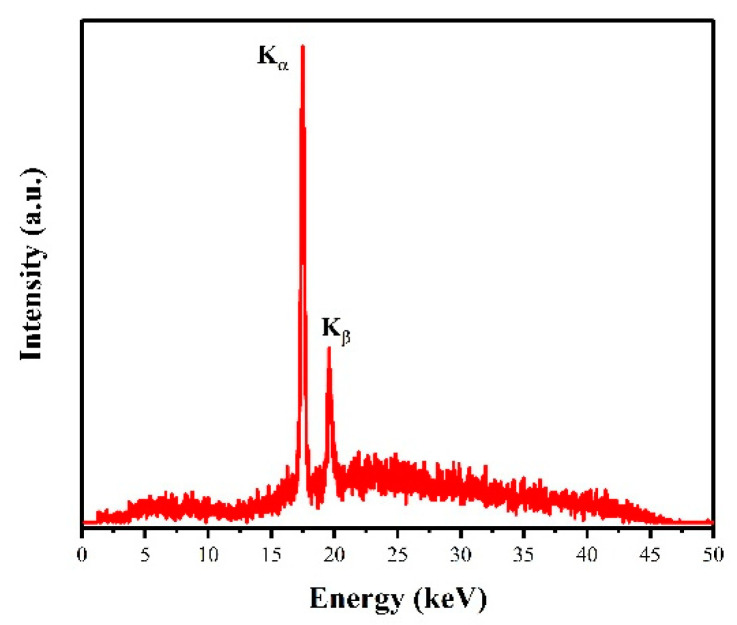
The X-ray spectrum of CNTs cold cathode X-ray source at V_a_ = 50 kV.

**Figure 9 nanomaterials-12-01882-f009:**
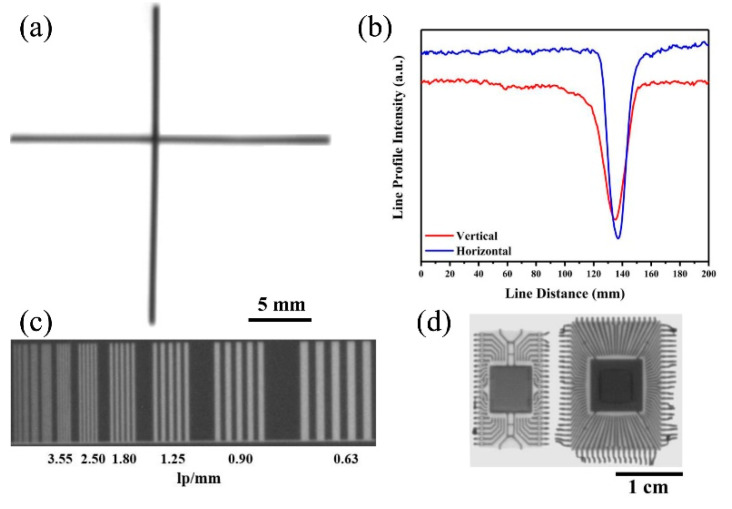
(**a**) X-ray projection images of vertically crossed tungsten wires. (**b**) Intensity profiles along the vertical and horizontal directions. X-ray images of line pair card (**c**) and integrated circuit chips (**d**).

**Table 1 nanomaterials-12-01882-t001:** The current stability of CNT electron sources reported in references.

CNT Preparation Method	Device Structure and Current Modulation Method	Gate Voltage	Cathode Current	Current Fluctuation	Ref.
Plasma enhanced CVD; Resist-assisted patterning	Gated structure without current modulation	1400 V	~6000 µA	2.4%	[30]
Electrophoretic deposition	Gated structure without current modulation	800 V	~600 µA	~1%	[44]
Microwave Plasma CVD	Diode structure with thin film transistor	N/A	~11 µA	<2%	[45]
Thermal CVD	Gated structure with CNTs on silicon posts as resistor ballast	60 V	~120 µA	<5%	[46]
Plasma Enhanced CVD; Resist-assisted patterning	Diode structure with MOSFET	N/A	~77 µA	~0.45%	[47]
Thermal CVD	Gated structure without current modulation	1200 V	~30 µA	3.91%	this work
Thermal CVD	Gated structure with IGBT	1200 V	15.3 µA	0.22%	this work

## Data Availability

Data are contained within the article.

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
