# Peer review of "Achieving High Current Stability of Gated Carbon Nanotube Cold Cathode Electron Source Using IGBT Modulation for X-ray Source Application"

_nanomaterials, 2022, doi:10.3390/nano12111882_

Round 1

Reviewer 1 Report

see attached report

Reviewer 2 Report

The paper "Achieving high current stability of gated carbon nanotube cold 2 cathode electron source using IGBT modulation for X-ray 3 source application" by Yajie Guo, Junfan Wang, Baohong Li, Yu Zhang, Shaozhi Deng and Jun Chen is of a very good quality.

They demonstrated the feasibility of achieving high current stability from the gated carbon nanotube cold cathode electron source using IGBT modulation for X-ray source application

The introduction is almost complete, however, I would suggest to add some references to consider also other important works in the field:

Line 34

"Since 2001, carbon nanotubes (CNTs) cold cathode has been extensively studied for 34 X-ray sources application due to its ultra-high aspect ratio and excellent electrical characteristics." Support the statement with a suitable reference. Here I suggest  https://doi.org/10.3390/app8040526

Line 65

"Several approaches have been studied for achieving high stable emission from 65 gated CNT cold cathode electron source for X-ray source application. Some researchers 66 optimized the preparation method or adopted a post-treatment process [27-28]." 

Stability of CNT cold cathodes is extensively discussed in the following paper https://doi.org/10.1016/j.carbon.2008.12.035, which could be added as a reference.

The device fabrication is convincingly well described.

The discussion of the results provided well includes characterizations and clear I-V curves.

The conclusions are fully credible and well justified.

I strongly recommend the publication in this journal.
